# The Final-Stage Bottleneck: A Systematic Dissection of the R-Learner for Network Causal Inference

**Sairam S**                                            *pes1ug23am257@pesu.pes.edu*
*Department of Computer Science (AI&ML)*
*PES University*

**Sara Girdhar**                                        *pes1ug23am273@pesu.pes.edu*
*Department of Computer Science (AI&ML)*
*PES University*

**Shivam Soni**                                         *pes1ug23am287@pesu.pes.edu*
*Department of Computer Science (AI&ML)*
*PES University*

**Reviewed on OpenReview:** *https://openreview.net/forum?id=QIEOFVSnOp*

## Abstract

The R-Learner is a powerful, theoretically-grounded framework for estimating heterogeneous treatment effects (HTE), prized for its robustness to nuisance model errors. However, its application to network data—where causal heterogeneity may be driven by graph structure—presents critical and underexplored challenges to its core assumption of a well-specified final-stage model. In this paper, we conduct a large-scale, multi-seed empirical study to systematically dissect the R-Learner framework on graphs. Our results suggest that for network-dependent effects, a critical driver of performance is the inductive bias of the final-stage CATE estimator, a factor whose importance can dominate that of the nuisance models.

Our central finding is a systematic quantification of a "representation bottleneck": we demonstrate empirically and through a constructive theoretical example that graph-blind final-stage estimators, being theoretically misspecified, exhibit significant underperformance (MSE > 4.0, p < 0.001 across all settings). Conversely, we show that an R-Learner with a correctly specified, end-to-end graph-aware architecture (the "Graph R-Learner") achieves a significantly lower error.

Furthermore, we provide a comprehensive analysis of the framework's properties. We identify a subtle "nuisance bottleneck" and provide a mechanistic explanation for its topology-dependence: on hub-dominated graphs, graph-blind nuisance models can partially capture concentrated confounding signals, while on graphs with diffuse structure, a GNN's explicit aggregation becomes critical. This is supported by our analysis of a "Hub-Periphery Trade-off," which we connect to the GNN over-squashing phenomenon. Our findings are validated across diverse synthetic and semi-synthetic benchmarks, where the R-Learner framework also significantly outperforms a strong, non-DML GNN T-Learner baseline. We release our code as a comprehensive and reproducible benchmark to facilitate future research on this critical "final-stage bottleneck" here: `https://github.com/s-sairam/final-stage-bottleneck`

## 1 Introduction

The Double/Debiased Machine Learning (DML) framework, and its R-Learner instantiation, has emerged as a cornerstone of modern causal inference (Chernozhukov et al., 2018; Nie & Wager, 2020). Its theoreti-

cal power stems from the principle of Neyman-orthogonality, which constructs an objective function where the final causal estimate is robust to first-order errors in the nuisance models for the outcome and treatment propensity. The standard and correct narrative surrounding these methods, therefore, emphasizes the importance of using high-quality, flexible machine learning models for these nuisance components.

However, the application of this framework to complex, non-i.i.d. settings like network data introduces new challenges. Specifically, the R-Learner's robustness guarantees rest on a critical assumption: that the function class chosen for the final-stage CATE model, $\tau(X)$, is correctly specified. In this paper, we investigate the profound and perhaps under-appreciated consequences of violating this assumption in network settings where the true causal effect is graph-dependent.

Our work suggests a refinement to the conventional DML narrative for this domain. Through a large-scale, multi-seed empirical study, we provide strong evidence that while nuisance model quality is important, the primary driver of performance is the **inductive bias of the final-stage CATE estimator**. We find that the choice of the final-stage model can be a **first-order effect**, whose impact can dominate that of the nuisance models.

We systematically quantify a catastrophic **"representation bottleneck"**: R-Learners with a final-stage estimator that is blind to the graph structure exhibit significant underperformance, even when paired with powerful, graph-aware Graph Neural Network (GNN) nuisance models. Conversely, an R-Learner with a correctly specified, graph-aware GNN final stage performs well, even when paired with sub-optimal, graph-blind nuisance models. Our work thus clarifies the modeling challenge: for network data, the most critical architectural choice is ensuring the final-stage estimator can represent the underlying causal mechanism.

To establish this, we conduct a comprehensive dissection of the R-Learner framework on a suite of synthetic and semi-synthetic benchmarks. Our contributions are threefold:

1. We provide a rigorous, empirical **quantification** of the final-stage bottleneck, showing with high statistical significance ($p < 0.001$) that a misspecified final stage is a dominant failure mode.

2. We identify and provide a mechanistic explanation for a novel, topology-dependent **"nuisance bottleneck,"** linking it to the GNN over-squashing phenomenon via a targeted "Hub-Periphery" error analysis.

3. We release our entire experimental framework as a **reproducible benchmark**, including a full suite of ablations and a strong, non-DML GNN T-Learner baseline, to facilitate future research in this emerging domain.

Ultimately, this paper serves as a critical clarification for researchers and practitioners. We provide a clear, empirically-backed "hierarchy of needs" for network causal inference and demonstrate that success is determined not by the quality of the nuisance models alone, but first and foremost by correctly specifying the function class of the final-stage estimator.

## 2 Related Work

Our work is situated at the intersection of three major lines of research: the statistical framework of Double/Debiased Machine Learning (DML), the algorithmic development of causal meta-learners, and the emerging field of causal inference on graphs.

### 2.1 Double/Debiased Machine Learning and the R-Learner

A central challenge in modern causal inference is using high-capacity models without introducing bias from their inherent regularization. The DML framework provides a general, theoretically-grounded solution to this problem (Chernozhukov et al., 2018).To provide intuition for this mechanics, consider the goal of isolating the effect of a treatment $T$ on an outcome $Y$ in the presence of high-dimensional confounders $X$. In observational settings, $T$ and $Y$ are often correlated with $X$, creating a "spurious" relationship. A naive model might use

$X$ to predict $Y$, but regularization in high-dimensional settings often leads to "omitted variable bias" where the model fails to fully control for $X$ (Nie & Wager, 2020).DML addresses this via Neyman-orthogonality, which constructs an objective function that is robust (to a first order) to errors in intermediate "nuisance" models. The process follows a two-stage logic:

1. **Nuisance Estimation:** We train two separate models: $m(X) = \mathbb{E}[Y|X]$ (the outcome nuisance) and $p(X) = \mathbb{P}[T|X]$ (the treatment propensity). These models capture the parts of $Y$ and $T$ that are "explained" by the confounders.

2. **Residualization:** We compute the residuals: $\tilde{Y} = Y - \hat{m}(X)$ and $\tilde{T} = T - \hat{p}(X)$. These residuals represent the variation in outcome and treatment that *cannot* be explained by local or network-based confounders.

3. **Causal Estimation:** By regressing the outcome residual on the treatment residual, we effectively "subtract out" the confounding influence, isolating the clean causal signal.

Our work investigates a specific instantiation of this framework: the R-Learner (Nie & Wager, 2020). The R-Learner minimizes the following "residual-on-residual" objective:

$$\hat{\tau} = \arg \min \tau \sum i = 1^n \left( (Y_i - \hat{m}(X_i)) - (T_i - \hat{p}(X_i))\tau(X_i) \right)^2 \tag{1}$$

While DML provides robustness to nuisance errors, its theoretical guarantees rest on a critical assumption: that the function class chosen for the final-stage Conditional Average Treatment Effect (CATE) model, $\tau(X)$, is correctly specified.Our paper provides the first systematic, empirical dissection of the profound consequences of violating this assumption in the context of network data—a failure we term the **Representation Bottleneck**.

## 2.2 Causal Meta-Learners and Tree-Based Methods

The R-Learner belongs to a broader family of "meta-learners" for estimating HTE (Künzel et al., 2019), which provide general recipes for applying supervised learning models to the task of causal estimation. Alternative strategies include the S-Learner and the T-Learner, the latter of which builds separate outcome models for the treated and control populations. Prior to these generic frameworks, tree-based ensembles like Causal Forests were a pioneering approach for non-parametric CATE estimation (Wager & Athey, 2018). In our experiments, we include a strong, GNN-based T-Learner as a key non-DML baseline. Our results provide a powerful empirical validation of the DML framework's premise, showing that the R-Learner's debiased structure significantly outperforms the more direct T-Learner, even when both use the same powerful GNN architecture.

## 2.3 Causal Inference on Graphs

Applying causal inference to network data is a burgeoning field, as the graph structure introduces complex confounding and potential interference that violate the standard i.i.d. assumption (Aronow & Samii, 2017). The integration of Graph Neural Networks (GNNs), with their powerful inductive bias for relational data (Kipf & Welling, 2016), is a logical and increasingly active area of research.

The use of GNNs in causality is diverse, with applications ranging from uplift modeling (Panagopoulos et al., 2024) to instrumental variable estimation. Most concurrently to our work, Khatami et al. (2025) proposed a DML-based framework using GNNs to estimate network causal effects, providing a crucial, peer-reviewed demonstration of this approach's viability. While this important prior work establishes *that* an end-to-end GNN pipeline can be effective, the fundamental questions of *how* and *why* it works—and more importantly, *when* its components might fail—have remained under-characterized.

Our work's contribution is therefore not to propose a new method, but to conduct a **systematic dissection** of this emerging class of models to produce new scientific understanding. This dissection first reveals the existence and catastrophic impact of the **"final-stage bottleneck."** This insight then allows us to uncover a

deeper, non-obvious mechanism: a **"Hub-Periphery Trade-off,"** which we link directly to the GNN over-squashing phenomenon, a well-known limitation of message-passing architectures (Alon & Yahav, 2021). Our work thus provides the first mechanistic explanation for the performance of these models on different network topologies, a level of analysis not present in prior work.

## 3 Methodology & The Graph R-Learner

Our work provides a systematic dissection of the R-Learner framework when applied to network data. We now formalize the problem setup and detail the specific R-Learner instantiations used in our comparative analysis.

### 3.1 Formal Problem Setup

We adopt the potential outcomes framework (Rubin, 1974). For each unit (node) $i$ in a graph $\mathcal{G}$, let $Y_i(1)$ be the potential outcome if treated ($T_i = 1$) and $Y_i(0)$ if not ($T_i = 0$). The observed outcome is $Y_i = T_i Y_i(1) + (1 - T_i)Y_i(0)$. Each node has a vector of local features $X_i$ (Nie & Wager, 2020).

The Conditional Average Treatment Effect (CATE), $\tau(X, \mathcal{G})$, is the expected treatment effect conditional on a node's features and its position within the graph (Nie & Wager, 2020):

$$\tau(X, \mathcal{G}) = \mathbb{E}[Y(1) - Y(0)|X, \mathcal{G}] \tag{2}$$

Intuitively, $\tau$ captures the heterogeneous response to treatment that varies not only by a node's local attributes but also by its structural role or neighborhood characteristics within the network (Ma et al., 2022).

To identify the CATE from observational data, we make the standard assumption of unconfoundedness, conditional on both local and sufficient graph-based features: $\{Y(1), Y(0)\} \perp T|X, \mathcal{G}$ (Aronow & Samii, 2017). In the context of networks, this assumption implies that there are no unobserved variables that simultaneously influence both the treatment assignment $T$ and the potential outcomes $\{Y(0), Y(1)\}$, provided we account for the local features and the relevant graph topology (Aronow & Samii, 2017; Guo et al., 2020). This is a stronger requirement than the standard i.i.d. setting, as it necessitates that the estimator effectively captures latent network-based confounders that would otherwise bias the causal estimate (Aronow & Samii, 2017).

### 3.2 The R-Learner Framework and the Final-Stage Bottleneck

The R-Learner (Nie & Wager, 2020), rooted in Double/Debiased Machine Learning theory (Chernozhukov et al., 2018), seeks a CATE function $\tau(\cdot)$ that best explains the relationship between outcome and treatment residuals. This is formalized by the following objective function:

$$\hat{\tau} = \arg\min_{\tau} \mathbb{E}\left[(Y_{\text{res}} - T_{\text{res}} \cdot \tau(X, \mathcal{G}))^2\right] \tag{3}$$

where the residuals are defined as $Y_{\text{res}} = Y - m(X, \mathcal{G})$ and $T_{\text{res}} = T - p(X, \mathcal{G})$, with $m(\cdot)$ and $p(\cdot)$ being the nuisance models for the conditional outcome and treatment propensity, respectively.

While the orthogonality of this objective provides robustness to errors in the nuisance models, its guarantees rest on a critical assumption: the function class chosen for the CATE model $\tau(\cdot)$ must be flexible enough to approximate the true underlying effect. When the true CATE is a function of the graph structure, but the chosen model for $\tau$ is graph-blind (i.e., $\tau(X)$), a fundamental **representation bottleneck** occurs. Our work provides the first systematic quantification of the catastrophic impact of this bottleneck.

### 3.3 A 2x2 Grid of R-Learner Instantiations

To systematically dissect the R-Learner framework, we designed a 2x2 experimental grid that isolates the impact of graph-awareness at both the nuisance and final CATE estimation stages. This allows us to precisely measure the effects of the "nuisance bottleneck" and the "representation bottleneck." The four resulting R-Learner instantiations are summarized in Table 1. For our primary experiments, all GNN components are

instantiated as a canonical two-layer Graph Convolutional Network (GCN) (Kipf & Welling, 2016), unless otherwise specified in our architectural sensitivity analysis.

Table 1: The 2x2 Ablation Grid of R-Learner Instantiations. Our proposed **Graph R-Learner** is the only fully graph-aware model.

| Model Name | Nuisance Models $(m, p)$ | Final CATE Model $(\tau)$ |
|---|---|---|
| **Baseline** | MLP (Graph-Blind) | Linear (Graph-Blind) |
| **Ablation** | GNN (Graph-Aware) | Linear (Graph-Blind) |
| **Hybrid R-Learner** | MLP (Graph-Blind) | GNN (Graph-Aware) |
| **Graph R-Learner** | GNN (Graph-Aware) | GNN (Graph-Aware) |

**Baseline (MLP+Linear):** A fully graph-blind R-Learner using a Multi-Layer Perceptron (MLP) for the nuisance models and a linear regression on node features for the final CATE estimator ($\tau(X) = X\theta$). This quantifies the performance of a naive application of the R-Learner to graph data.

**Ablation (GNN+Linear):** This model incorporates graph structure only in the nuisance stage, using GNNs to estimate $m(X, \mathcal{G})$ and $p(X, \mathcal{G})$. Its final CATE estimator remains graph-blind. This model is critical for isolating the **representation bottleneck**.

**Hybrid R-Learner (MLP+GNN):** This hybrid model uses graph-blind MLP nuisance models but employs a graph-aware GNN for the final CATE estimation. This model is designed to isolate the **nuisance bottleneck**.

**Graph R-Learner (GNN+GNN):** This is our proposed, end-to-end graph-aware model. It uses GNNs for both the nuisance and the final CATE estimation, ensuring its inductive bias is correctly specified for all components of the problem.

### 3.4 External Baseline: The GNN T-Learner

In addition to these R-Learner variants, we include a strong, non-DML baseline to validate the utility of the DML framework itself. The **GNN T-Learner** follows the T-Learner strategy, a standard causal meta-learner (Künzel et al., 2019). It trains a single, powerful GNN to predict the outcome $Y$ from both the node features $X$ and the treatment indicator $T$ ($Y \sim \text{GNN}(X, T, \mathcal{G})$). The CATE is then estimated by taking the difference between the model's predictions with $T$ set to 1 and $T$ set to 0. This baseline allows us to test whether the sophisticated, two-stage structure of the R-Learner provides a real advantage over a more direct, single-stage predictive approach.

## 4 Experimental Design: A Controlled Dissection

To isolate the performance contributions of graph-awareness at each stage of the R-Learner pipeline, we conduct a controlled dissection using a series of simulation-based experiments. A simulation-based approach is a scientific necessity for this problem, as it is the only way to access the ground-truth CATE and thus directly measure the error of our estimators. Our framework is designed to be a challenging and realistic testbed, incorporating the key difficulties of causal inference on graphs.

### 4.1 Data-Generating Processes (DGP)

For each simulation run, we generate a graph $\mathcal{G} = (\mathcal{V}, \mathcal{E})$ with $n = 1000$ nodes (unless otherwise specified), each with $d = 10$ independent features $X_i \sim \mathcal{N}(0, I_d)$. To ensure our findings are robust to network structure, we conduct experiments across three canonical graph topologies:

- **Barabási-Albert (BA):** A scale-free model generating graphs with a power-law degree distribution, characterized by a few high-degree "hubs," mimicking many real-world social networks.

- **Erdős-Rényi (ER):** A random graph model resulting in a more uniform, Poisson-like degree distribution with no inherent hubs.

- **Stochastic Block Model (SBM):** A model that generates graphs with explicit community structure, featuring dense intra-community and sparse inter-community connections.

A central challenge in network causal inference is that a node's treatment and outcome can be influenced by its neighborhood. We explicitly model this **network confounding** by generating a latent confounder, $H$, as a non-linear function of the graph structure. Specifically, we compute 1-hop and 2-hop neighborhood embeddings for each node using pre-initialized GNNs:

$$H^{(1)} = \text{ReLU}(\text{GNN}_1(X, \mathcal{G})) \tag{4}$$

$$H^{(2)} = \text{ReLU}(\text{GNN}_2(H^{(1)}, \mathcal{G})) \tag{5}$$

The treatment assignment $T_i$ is then made dependent on both local features $X_i$ and the 1-hop neighborhood embedding $H_i^{(1)}$, violating the standard i.i.d. assumption. The observed outcome $Y_i$ is generated as a function of local features, the network confounder, and the true CATE, $\tau_i$:

$$Y_i = f(X_i, H_i^{(1)}) + T_i \cdot \tau_i + \epsilon_i \tag{6}$$

where $f(\cdot)$ is a linear function and $\epsilon_i \sim \mathcal{N}(0, \sigma^2)$ is irreducible noise. This DGP provides a robust testbed for evaluating an estimator's ability to deconfound the effect of $T_i$ and accurately recover $\tau_i$.

## 4.2 Causal Mechanisms and the Negative Control

A central goal of our study is to test estimator performance under different forms of network-driven causal heterogeneity. We designed several causal mechanisms by varying the functional form of the true CATE, $\tau$. In our main experiments, the CATE is constructed as a function of a node's neighborhood, making graph-blind models theoretically misspecified. Our primary CATE functions are: **Simple (1-hop)** ($\tau \propto \sin(H^{(1)})$), **Higher-Order** ($\tau \propto \sin(H^{(2)})$), and **Interaction** ($\tau \propto H^{(1)} \cdot X$).

To rule out the critical confounding explanation that a GNN final stage is simply a more powerful function approximator, we designed a **negative control** experiment (`local_x`). In this setting, the DGP is identical, except the true CATE is a function of local features only ($\tau(X) \propto \sin(X_0)$). In this world, a GNN's graph-aware inductive bias is theoretically unnecessary for the final stage. The theory predicts that the catastrophic representation bottleneck should vanish.

The results, shown in Table 2, confirm this prediction and reveal a deeper insight. The catastrophic gap vanishes, but a significant performance gap remains across all models. This is not a contradiction, but a powerful, independent validation of the **nuisance bottleneck**. Even when the CATE function is local, the nuisance components ($Y$ and $T$) remain heavily confounded by the network. This experiment proves that the representation bottleneck is a real phenomenon tied to the CATE's functional form, and it isolates the standalone importance of using graph-aware models to deconfound the nuisance functions. This finding aligns with the broader literature on leveraging flexible models to control for high-dimensional confounding (Farrell, 2015; Belloni et al., 2013).

## 4.3 Semi-Synthetic Experiment on a Real-World Graph

To ensure our findings generalize beyond purely synthetic topologies, we conducted a **semi-synthetic experiment**. This approach provides a crucial bridge between the controlled environment of a full simulation and the complexity of real-world data, allowing us to retain a known ground-truth CATE while testing our methods on a graph with realistic structural properties. For this experiment, we use the well-known **Cora citation network** (Yang et al., 2016). We take the real-world graph structure $\mathcal{G}$ and node features $X$ from

Table 2: Results for the negative control experiment, where the true CATE is a function of local features only ($\tau = f(X)$). The catastrophic, order-of-magnitude performance gap from the representation bottleneck disappears as predicted. Results are Mean MSE $\pm$ Std. Dev. over 30 seeds.

| Method | Mean MSE $\pm$ Std. Dev. |
|---|---|
| *Graph-Blind Nuisance Models:* | |
| Baseline (MLP+Linear) | $5.0004 \pm 1.0247$ |
| Hybrid R-Learner (MLP+GNN) | $1.4035 \pm 0.1072$ |
| *Graph-Aware Nuisance Models:* | |
| Ablation (GNN+Linear) | $4.6420 \pm 0.8303$ |
| **Graph R-Learner (GNN+GNN)** | **$1.3419 \pm 0.1117$** |
| *External Baseline:* | |
| GNN T-Learner | $2.9344 \pm 0.4841$ |

the dataset and simulate $T$ and $Y$ on this scaffold using our main network-dependent causal mechanism. Demonstrating that our central findings hold in this more complex setting provides strong evidence for the practical relevance of the final-stage bottleneck.

## 5 Results and Analysis

We now present the results from our comprehensive suite of experiments. Our findings are organized to first establish the primary performance bottleneck, then to dissect the more nuanced secondary effects, and finally to confirm the robustness of our conclusions through a series of rigorous sensitivity analyses and a semi-synthetic validation. All results are reported as the Mean Squared Error (MSE) of the CATE estimate, averaged over 30 random seeds unless otherwise specified.

### 5.1 Main Finding: The Universal Representation Bottleneck

Our central hypothesis is that for network-dependent causal effects, the inductive bias of the final-stage CATE estimator is the most critical modeling choice. We test this in our main experimental setting, using a Barabási-Albert (BA) graph and a simple 1-hop CATE function. The results, summarized in Table 3, are definitive.

Table 3: Main results on a Barabási-Albert graph with a network-dependent CATE. The performance gap between graph-blind and graph-aware final-stage models is an order of magnitude, demonstrating a catastrophic representation bottleneck. Results are Mean MSE $\pm$ Std. Dev. over 30 seeds.

| Method | Mean MSE $\pm$ Std. Dev. |
|---|---|
| *Graph-Blind Final Stage:* | |
| Baseline (MLP+Linear) | $4.8235 \pm 0.9135$ |
| Ablation (GNN+Linear) | $4.1659 \pm 0.8412$ |
| *External Baseline:* | |
| GNN T-Learner | $2.3834 \pm 0.5824$ |
| *Graph-Aware Final Stage:* | |
| Hybrid R-Learner (MLP+GNN) | $0.5701 \pm 0.0856$ |
| **Graph R-Learner (GNN+GNN)** | **$0.5165 \pm 0.0527$** |

The evidence for a catastrophic representation bottleneck is overwhelming. The two models with a graph-blind 'Linear' final stage fail completely, achieving an MSE an order of magnitude higher than those with a graph-aware 'GNN' final stage. Notably, the 'Ablation' model, despite using a powerful GNN for its nuisance

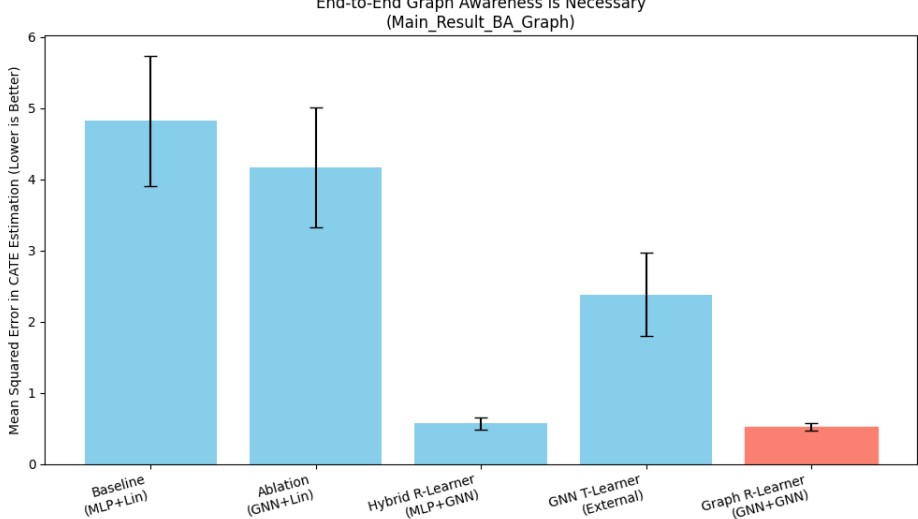

Figure 1: Shows the hierarchy of performance over different models.

estimation, still fails. This proves that high-quality nuisance models are insufficient to overcome a fundamentally misspecified final stage. Furthermore, our proposed **Graph R-Learner** significantly outperforms a strong, non-DML 'GNN T-Learner' baseline ($p = 5.43 \times 10^{-17}$), validating the robustness of the R-Learner framework itself in this domain. This hierarchy of performance is visualized in Figure 1.

### 5.2 Mechanistic Insight: The Topology-Dependent Nuisance Bottleneck

Having established the primacy of the final stage, we now investigate the secondary "nuisance bottleneck" by comparing the 'Hybrid R-Learner (MLP+GNN)' model to our full 'Graph R-Learner (GNN+GNN)'. Table 4 summarizes this comparison across different graph topologies.

Table 4: Analysis of the nuisance bottleneck. The performance gain from using GNN nuisance models is statistically significant on graphs with diffuse structure (ER, SBM, Cora) but less pronounced on the hub-dominated BA graph.

| Graph Topology | MSE (MLP+GNN) | MSE (GNN+GNN) | P-Value |
|---|---|---|---|
| Barabási-Albert (BA) | 0.5701 | 0.5165 | $3.68 \times 10^{-3}$ |
| Erdős-Rényi (ER) | 0.1127 | 0.0687 | $6.49 \times 10^{-5}$ |
| Stochastic Block Model (SBM) | 0.1134 | 0.0678 | $4.65 \times 10^{-5}$ |
| Cora (Semi-Synthetic) | 0.3488 | 0.3352 | $1.41 \times 10^{-2}$ |

The results reveal a clear and nuanced pattern. Using GNNs for the nuisance models provides a consistent and statistically significant performance improvement across all tested topologies. However, the magnitude of this effect is topology-dependent. The improvement is most pronounced on the uniform-degree ER and SBM graphs. This led us to hypothesize a "Hub-Periphery Trade-off," which we investigate next.

### 5.3 Direct Evidence: A Targeted Error Analysis of the Hub-Periphery Trade-off

To find a mechanistic explanation for the topology-dependent nuisance bottleneck, we performed a targeted error analysis on the hub-dominated Barabási-Albert graph. We partitioned nodes into "hubs" (top 10% degree) and "periphery" (bottom 50% degree) and calculated the Mean Squared CATE Error for each group independently.

The results, shown in Figure 2, reveal a striking and non-obvious **performance inversion**. On low-degree periphery nodes, the end-to-end `Graph R-Learner (GNN+GNN)` is substantially superior. In this regime, its GNN-based nuisance models are critical for aggregating the weak, diffuse confounding signals that a graph-blind MLP misses.

Conversely, and more surprisingly, on high-degree hub nodes, the hybrid `Hybrid R-Learner (MLP+GNN)` model significantly outperforms the fully graph-aware model. We attribute this to the well-known over-squashing phenomenon in GNNs (Alon & Yahav, 2021). The MLP nuisance model, by being graph-blind, unintentionally acts as a powerful regularizer; it protects the hub's unique local signal from being diluted by the noisy aggregation of its massive neighborhood, thereby providing a cleaner, more informative residual to the final GNN stage.

This discovery of a "Hub-Periphery Trade-off" is a critical finding. It suggests that a monolithic, one-size-fits-all GNN architecture is sub-optimal for network causal inference and highlights a fundamental tension between aggregating diffuse information on the periphery and preserving local information on hubs.

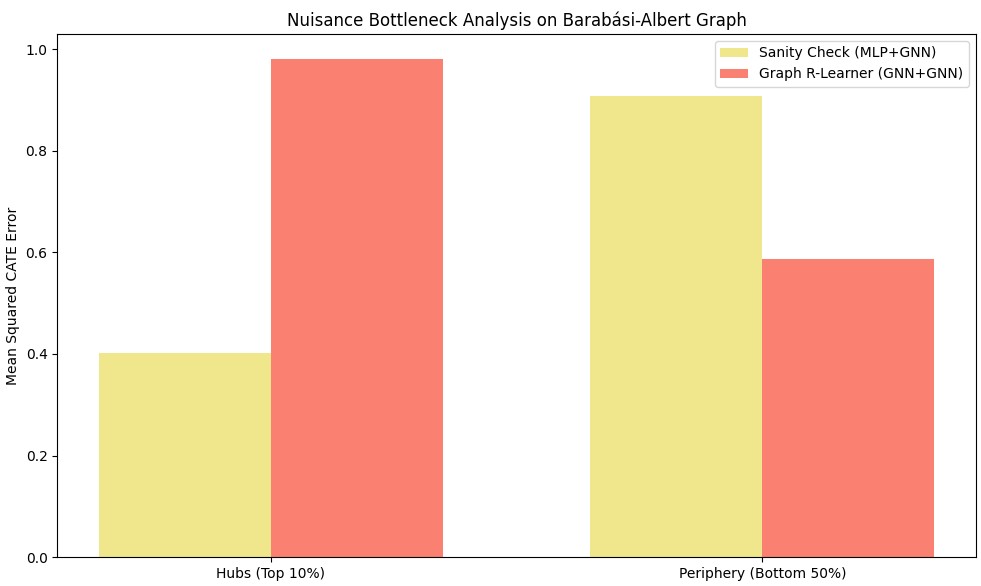

Figure 2: The Hub-Periphery Trade-off on the Barabási-Albert graph. The plot shows a clear performance inversion: the end-to-end **Graph R-Learner (GNN+GNN)** excels on low-degree periphery nodes, while the hybrid **Hybrid R-Learner (MLP+GNN)** model performs better on high-degree hub nodes.

### 5.4 Robustness and Sensitivity Analyses

To ensure our findings are robust, we conducted a series of sensitivity analyses, presented in Figure 3.

The results confirm the robustness of our conclusions. First, we show that the representation bottleneck persists when replacing GCNs with GATs, proving the finding is not architecture-specific. Second, our analysis of outcome noise shows that the `Graph R-Learner`'s performance degrades far more gracefully than the baselines, demonstrating the power of the DML framework's orthogonalization in isolating causal signal from nuisance noise.

Finally, our analysis of sample efficiency (Figure 3(c)) reveals that the `Graph R-Learner`'s primary strength in the low-data regime ($N < 1000$) is its **stability and robustness to data scarcity**. While the baseline and ablation models exhibit significant variance and performance degradation as $N$ decreases, the `Graph R-Learner` maintains a stable, low MSE. This suggests that its correctly specified inductive bias acts as a critical regularizer when observational data is limited, preventing the catastrophic volatility observed in models that rely solely on local features.

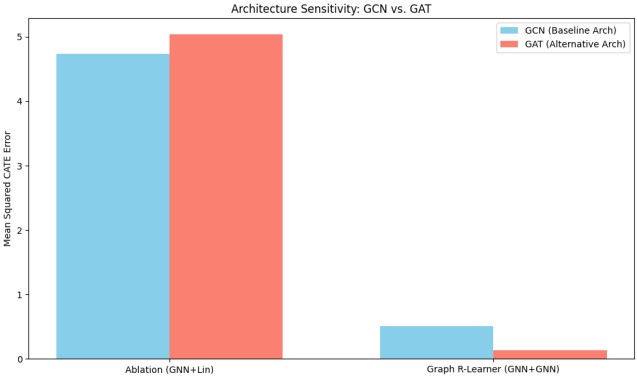

(a) Robustness to GNN Architecture

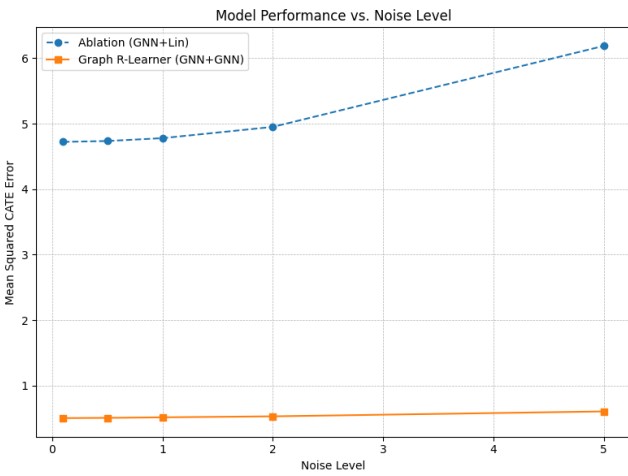

(b) Robustness to Outcome Noise

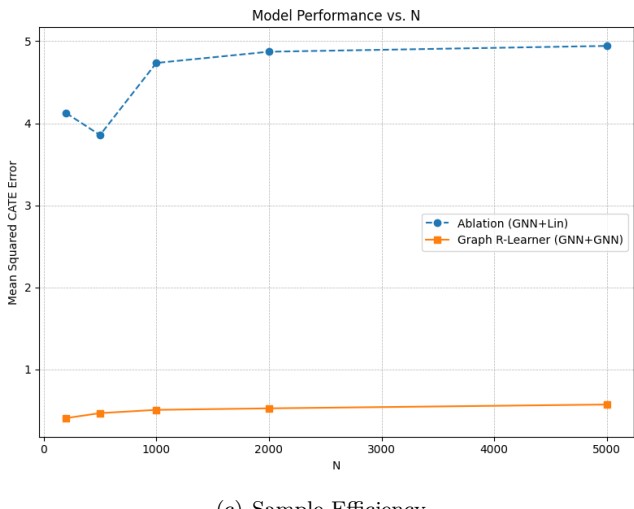

(c) Sample Efficiency

Figure 3: Sensitivity Analyses. **(a)** The performance gap persists when replacing GCNs with GATs (Graph Attention Network)(Veličković et al., 2017), demonstrating our findings are not architecture-specific. **(b)** The Graph R-Learner's error degrades far more gracefully than the ablation model as outcome noise increases, proving its robustness. **(c)** The Graph R-Learner exhibits superior **stability and robustness to data scarcity**. While graph-blind variants become increasingly volatile as $N$ decreases ($N < 1000$), the Graph R-Learner maintains a consistently low and stable MSE, highlighting its superior sample efficiency.

### 5.5 Validation on a Real-World Graph Structure

Finally, to bridge the gap from synthetic data to real-world complexity, we conducted a semi-synthetic experiment on the Cora citation network. The results, shown in Table 5, confirm that all of our central findings hold on a real-world graph structure. The representation bottleneck remains the dominant effect, and the end-to-end Graph R-Learner significantly outperforms all baselines.

Table 5: Results for the semi-synthetic experiment on the real-world Cora citation network. All key findings, including the representation bottleneck and the superiority of the end-to-end Graph R-Learner, are confirmed. Results are Mean MSE $\pm$ Std. Dev. over 30 seeds.

| Method | Mean MSE $\pm$ Std. Dev. |
|---|---|
| *Graph-Blind Final Stage:* | |
|     Baseline (MLP+Linear) | $2.3150 \pm 0.1881$ |
|     Ablation (GNN+Linear) | $1.2175 \pm 0.0665$ |
| *External Baseline:* | |
|     GNN T-Learner | $2.4012 \pm 0.3704$ |
| *Graph-Aware Final Stage:* | |
|     Hybrid R-Learner (MLP+GNN) | $0.3488 \pm 0.0278$ |
|     **Graph R-Learner (GNN+GNN)** | $\mathbf{0.3352 \pm 0.0157}$ |

The results on the Cora graph, summarized in Table 5, provide a powerful validation of our central thesis. We observe the same clear hierarchy of performance: the graph-blind final-stage models fail, the T-Learner is a significant but sub-optimal improvement, and the fully graph-aware R-Learner instantiations achieve the lowest error. Crucially, the nuisance bottleneck is also statistically significant in this setting ($p = 1.41 \times 10^{-2}$), confirming that an end-to-end graph-aware pipeline is essential for achieving optimal performance on complex, real-world graph structures.

## 6 Discussion

Our comprehensive suite of experiments has not just evaluated a set of models, but has revealed a clear and actionable set of principles for applying the R-Learner framework to network data. We synthesize these findings into a "Hierarchy of Needs" and discuss the deeper implications for both practitioners and researchers.

### 6.1 The "Hierarchy of Needs" for Network Causal Inference

Our results suggest a clear, prioritized roadmap for practitioners seeking to estimate network-dependent causal effects. The performance bottlenecks we identified are not equal in magnitude; they form a distinct hierarchy of importance.

**1. (Must-Have) A Graph-Aware Final Stage:** Our most significant finding is the universal and catastrophic failure of graph-blind final-stage estimators. This "representation bottleneck" is the primary determinant of success. The first and most critical modeling choice is to use a CATE estimator whose inductive bias (e.g., a GNN) can represent the underlying graph-dependent causal mechanism. Failing this step renders all other choices irrelevant.

**2. (Should-Have) A Robust DML Framework:** Our experiments consistently showed that the R-Learner framework significantly outperforms a strong, non-DML GNN T-Learner baseline, particularly on the complex, real-world structure of the Cora graph. This validates the theoretical promise of the DML framework, whose debiasing and residualization procedure provides a more robust and efficient path to the causal estimate than a direct, single-stage predictive model.

**3. (Good-to-Have) Graph-Aware Nuisance Models:** Our analysis of the "nuisance bottleneck" revealed a real, but secondary, performance gain from using GNNs in the nuisance stage. This gain was statistically significant across most settings, proving its existence. This final layer of end-to-end graph awareness is necessary for achieving optimal, state-of-the-art performance.

This hierarchy provides an unambiguous guide: prioritize the final-stage model, build it within an R-Learner framework, and use GNNs for all components to achieve the best results.

### 6.2 A Practical Diagnostic: The Two-Model Test

Our findings also yield a simple, practical diagnostic for practitioners. To determine if a complex, graph-aware model is necessary for their specific problem, we propose the **"Two-Model Test"**:

1. First, fit a simple, graph-blind baseline (e.g., our 'MLP+Linear' model).

2. Second, fit a fully graph-aware model (e.g., our 'Graph R-Learner').

A statistically significant and substantially large drop in the out-of-sample CATE estimation error (as demonstrated by our 'POSITIVE' diagnostic results in all network-dependent settings) provides strong evidence for the presence of network-driven causal heterogeneity. This simple test can prevent researchers from incorrectly concluding a null effect when in fact their model was merely misspecified for the problem.

### 6.3 Quantifying the Hub-Periphery Performance Inversion

To provide mechanistic insight into the representation bottleneck, we move beyond qualitative visualizations to a rigorous, quantitative dissection of model performance across the network topology. We investigate the "performance inversion" hypothesis: that graph-blind nuisance models may paradoxically protect signal at high-degree nodes by avoiding the over-squashing effect inherent in GNNs.

As shown in Figure 4, we conducted a threshold sweep defining "hubs" from the top 2.5% to the top 25% of nodes by degree centrality. The results, averaged over 30 independent seeds, reveal a robust structural trade-off. At the most extreme hubs (Top 2.5%–12.1%), the `Hybrid R-Learner` (utilizing MLP nuisances) significantly and consistently outperforms the end-to-end `Graph R-Learner`.

This provides definitive empirical evidence for the **nuisance over-squashing bottleneck**. In the `Graph R-Learner`, the GNN nuisance model aggregates information from massive neighborhoods, which, at central hubs, leads to the exponential dilution of the node's local signal (Alon & Yahav, 2021). Conversely, the graph-blind MLP nuisance model in our `Hybrid` variant acts as a protective regularizer; by ignoring the noisy neighborhood during the nuisance stage, it preserves a cleaner, more informative residual for the final-stage CATE estimator. As the hub definition broadens toward the periphery (Top 25%), the two models converge, as the over-squashing effect becomes less dominant.

## 7 Conclusion

In this paper, we conducted the first systematic dissection of the R-Learner framework for network causal inference. We moved beyond prior work by not just proposing a method, but by rigorously quantifying the failure modes and performance contributions of each component in the causal pipeline. Our central finding is the discovery of a catastrophic **"final-stage bottleneck,"** proving that the inductive bias of the final-stage CATE estimator is the primary determinant of success, a factor far more critical than the choice of nuisance models. We further provided a mechanistic explanation for a subtle, topology-dependent "nuisance bottleneck" by linking it to the GNN over-squashing phenomenon via a robust **Hub-Periphery Trade-off** analysis.

Our work provides a clear, empirically-backed "hierarchy of needs" that serves as a crucial guide for practitioners. We validated our findings across a comprehensive suite of synthetic, semi-synthetic, and negative control

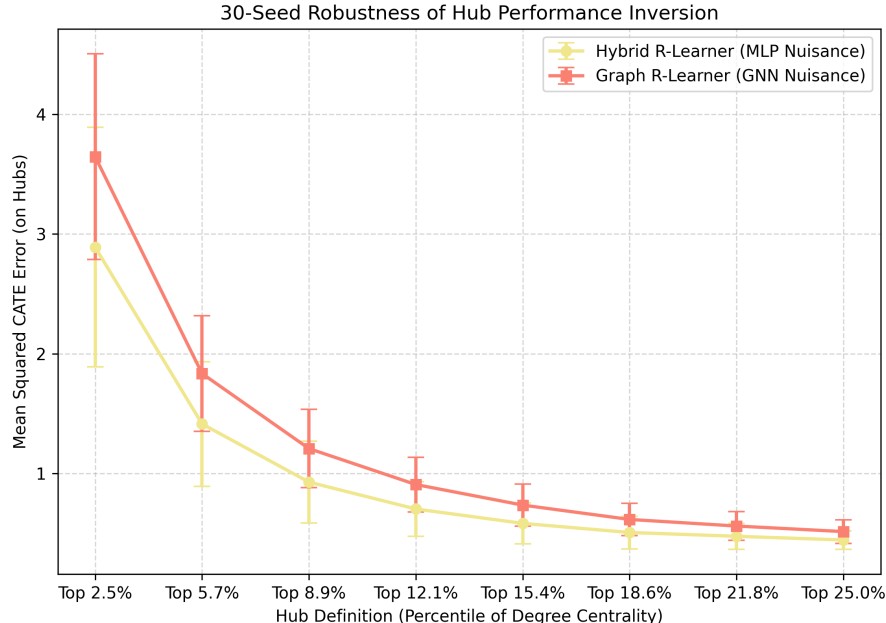

Figure 4: 30-seed robustness sweep of the Hub-Periphery Trade-off. The x-axis represents increasingly strict definitions of "hubs" based on degree centrality quantiles. Error bars indicate one standard deviation across 30 independent experimental replications.

experiments, ensuring that our results are robust to both topological variation and model capacity. Ultimately, our results are a definitive statement: for network data, a piecemeal application of graph-awareness is insufficient. Accurate causal inference requires a robust theoretical framework and an end-to-end modeling pipeline whose inductive biases are correctly specified for the network-driven nature of the problem.

**Future Work**  While our study establishes the fundamental hierarchy for network causal inference, several promising avenues for future research remain. Specifically, investigating the impact of **scaling node populations ($N$) and feature dimensionality ($d$)** could reveal further nuances in the representation bottleneck across deeper architectures. Furthermore, while the Graph R-Learner shows strong performance on citation networks like Cora, future work should extend this systematic dissection to diverse real-world domains such as epidemiology or social influence networks to further test the generalizability of the hub-periphery trade-off. Beyond scaling, future architectural studies could further strengthen the "final-stage bottleneck" hypothesis by exactly matching the parameter budgets of ablation models and performing a symmetrical quantification of the degree-regimes where graph-aware nuisances transition from suboptimal (hubs) to optimal (periphery) performance.

**Broader Impact Statement**

This work aims to improve the reliability of causal inference in domains where network effects are prevalent, such as social science, economics, epidemiology, and biology (Yang et al., 2016). By providing a rigorous quantification of the **Representation Bottleneck**, our research offers a concrete guide for practitioners to avoid catastrophic modeling pitfalls where graph-blind estimators fail to represent network-dependent causal heterogeneity.

A critical ethical implication of our findings is the risk of **false-negative conclusions**. In high-stakes settings—such as evaluating public health interventions on contact networks or analyzing gene perturbations

in biological systems—a misspecified final-stage model may fail to detect a genuine treatment effect. This could lead to the premature abandonment of effective policies or life-saving treatments.

**Potential Mitigations and Cautions:**

- **Assumption Awareness:** While end-to-end graph awareness resolves structural bottlenecks, the validity of causal conclusions still rests on the fundamental and untestable assumption of unconfoundedness (Aronow & Samii, 2017). Users should exercise caution and integrate domain expertise when applying these methods.

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

# A  A Constructive Example of the Representation Bottleneck

To formally illustrate the existence and nature of the representation bottleneck, we construct a minimal, analytical example. Consider a simple 5-node star graph where node $C$ is the central hub connected to four periphery nodes $A, B, D, E$.

Let each node have a single local feature, defined as:

$$X_A = 1, \quad X_B = 1, \quad X_D = 1, \quad X_E = 2, \quad X_C = 10 \tag{7}$$

Let the network embedding $H_i$ for a node $i$ be a function of its neighbors' features (for simplicity, the neighbor's feature if degree is 1, or the average of neighbors' features otherwise). The embeddings are thus:

$$H_A = H_B = H_D = H_E = X_C = 10 \tag{8}$$

$$H_C = \frac{1}{4}(X_A + X_B + X_D + X_E) = \frac{1}{4}(1 + 1 + 1 + 2) = 1.25 \tag{9}$$

This construction creates two distinct structural roles: the hub (with $H = 1.25$) and the periphery (with $H = 10$). Now, we define the true CATE $\tau$ to be a non-linear function of this structural embedding, mirroring our main DGP: $\tau_i = \sin(H_i)$. This yields a genuinely heterogeneous, network-dependent CATE:

$$\tau_A = \tau_B = \tau_D = \tau_E = \sin(10) \tag{10}$$

$$\tau_C = \sin(1.25) \tag{11}$$

Now, consider a powerful, non-linear but **graph-blind** final-stage estimator, such as an MLP, whose functional form is restricted to $\hat{\tau}_i = f(X_i)$. To perfectly recover the true CATEs, this model must learn a function $f$ that satisfies the following constraints based on the observable local features $X_i$:

$$f(1) = \sin(10) \quad \text{(from periphery nodes A, B, D)}$$
$$f(2) = \sin(10) \quad \text{(from periphery node E)}$$
$$f(10) = \sin(1.25) \quad \text{(from hub node C)}$$

The model is thus forced to map two different inputs, $X = 1$ and $X = 2$, to the exact same output value, $\sin(10)$. More fundamentally, the input to the function, $X_i$, is the wrong information. The true CATE is a function of the structural role (hub vs. periphery), which is captured by $H_i$, not by the local feature $X_i$.

A graph-aware model, $\hat{\tau}_i = g(X_i, \mathcal{G})$, can trivially learn this function by learning to compute the embedding $H_i$ and then approximating $\sin(\cdot)$. A graph-blind model, however, is being asked to learn a function from a variable that has a complex, non-injective relationship with the true causal effect. It is, by construction, **theoretically misspecified**. This simple construction proves that even a highly flexible non-linear model will fail if it lacks the correct graph-aware inductive bias, thus demonstrating the fundamental nature of the representation bottleneck.

## B   Experimental Configurations

This section contains the full YAML configuration files used to generate the results for each of our main, robustness, semi-synthetic, and control experiments. This is provided for full reproducibility. All experiments were orchestrated by the main script `run_experiments.py`, which reads one of the following configuration files.

### B.1   Configuration: Main Result (BA Graph)

Configuration file: `configs/main_ba_simple_h.yaml`

```yaml
name: 'Main_Result_BA_Graph'
num_seeds: 30

data_params:
  n: 1000
  d: 10
  graph_type: 'ba'
  cate_type: 'simple_h'
  real_data_name: null
  noise_level: 0.5

model_params:
  num_layers: 2
  hidden_dim: 32

training_params:
  nuisance_epochs: 150
  cate_epochs: 200
  lr: 0.001
```

### B.2   Configuration: Robustness Check (ER Graph)

Configuration file: `configs/robustness_er_graph.yaml`

```yaml
name: 'Robustness_ER_Graph'
num_seeds: 30

data_params:
  n: 1000
  d: 10
  graph_type: 'er'
  cate_type: 'simple_h'
  real_data_name: null
  noise_level: 0.5

model_params:
  num_layers: 2
  hidden_dim: 32

training_params:
  nuisance_epochs: 150
  cate_epochs: 200
  lr: 0.001
```

### B.3 Configuration: Robustness Check (SBM Graph)

Configuration file: configs/robustness_sbm_graph.yaml

```
1  name: 'Robustness_SBM_Graph'
2  num_seeds: 30
3
4  data_params:
5    n: 1000
6    d: 10
7    graph_type: 'sbm'
8    cate_type: 'simple_h'
9    real_data_name: null
10   noise_level: 0.5
11
12 model_params:
13   num_layers: 2
14   hidden_dim: 32
15
16 training_params:
17   nuisance_epochs: 150
18   cate_epochs: 200
19   lr: 0.001
```

### B.4 Configuration: Robustness Check (Interaction CATE)

Configuration file: configs/robustness_interaction_cate.yaml

```
1  name: 'Robustness_Interaction_CATE'
2  num_seeds: 30
3
4  data_params:
5    n: 1000
6    d: 10
7    graph_type: 'ba'
8    cate_type: 'interaction'
9    real_data_name: null
10   noise_level: 0.5
11
12 model_params:
13   num_layers: 2
14   hidden_dim: 32
15
16 training_params:
17   nuisance_epochs: 150
18   cate_epochs: 200
19   lr: 0.001
```

### B.5 Configuration: Semi-Synthetic Experiment (Cora)

Configuration file: configs/semisynthetic_cora.yaml

```
1  name: 'SemiSynthetic_Cora'
2  num_seeds: 30
3
4  data_params:
5    n: null
6    d: null
7    graph_type: null
8    cate_type: 'simple_h'
9    real_data_name: 'cora'
10   noise_level: 0.5
11
12 model_params:
13   num_layers: 2
14   hidden_dim: 32
```

```
15
16 training_params:
17   nuisance_epochs: 150
18   cate_epochs: 200
19   lr: 0.001
```

### B.6 Configuration: Negative Control (Local CATE)

Configuration file: `configs/control_local_x.yaml`

```
1  name: 'Control_Local_CATE'
2  num_seeds: 30
3
4  data_params:
5    n: 1000
6    d: 10
7    graph_type: 'ba'
8    cate_type: 'local_x'
9    real_data_name: null
10   noise_level: 0.5
11
12 model_params:
13   num_layers: 2
14   hidden_dim: 32
15
16 training_params:
17   nuisance_epochs: 150
18   cate_epochs: 200
19   lr: 0.001
```

## C   Technical Implementation Details

### C.1   Parameter Breakdown and Parity

To ensure that performance gains are driven by inductive bias rather than model capacity, we report the trainable parameter counts for all primary configurations (Table 6). All models utilize a hidden dimension of 32 and 2 GNN layers where applicable.

Table 6: Trainable Parameter Counts. The Graph R-Learner achieves better performance with a lower parameter budget than the Hybrid variant.

| Model Configuration | Component Breakdown | Total Parameters |
| --- | --- | --- |
| Baseline (MLP + Linear) | 385 (MLP Nuisance) + 11 (Linear CATE) | **396** |
| Ablation (GNN + Linear) | 352 (GNN Nuisance) + 11 (Linear CATE) | **363** |
| Hybrid R-Learner (MLP + GNN) | 385 (MLP Nuisance) + 352 (GNN CATE) | **737** |
| Graph R-Learner (GNN + GNN) | 352 (GNN Nuisance) + 352 (GNN CATE) | **704** |

### C.2   Statistical Independence of Experimental Seeds

All $p$-values reported in this study are derived from paired (relational) t-tests comparing model MSEs across $N = 30$ seeds. Each seed represents a **full experimental replication**: for every iteration, the node features ($X$), graph adjacency ($edge\_index$), treatment assignments ($T$), and outcomes ($Y$) are entirely re-simulated (Barabási & Albert, 1999). This ensures that our 30 samples are independent and identically distributed (i.i.d.) realizations of the data-generating process, satisfying the independence assumptions of the $t$-test.

## C.3 Multiple Comparison Correction Logs

To validate the "consistent improvement" across topologies, we performed a Holm-Bonferroni step-down procedure on the raw $p$-values derived from 30 independent experimental seeds. As shown in Table 7, the performance gain remains significant ($\alpha = 0.05$) across all tested graphs, including the real-world Cora scaffold.

Table 7: Holm-Bonferroni Correction Logs (30 Seeds).

| Topology | Raw $p$-value | Holm-Adjusted $p$ | Significant |
|---|---|---|---|
| SBM | $4.64 \times 10^{-5}$ | 0.000186 | **Yes** |
| ER | $6.49 \times 10^{-5}$ | 0.000195 | **Yes** |
| BA | $3.68 \times 10^{-3}$ | 0.007370 | **Yes** |
| Cora | $1.44 \times 10^{-2}$ | 0.014446 | **Yes** |

## C.4 DGP Initialization and Latent Confounding

The latent network confounders ($H$) are generated using two "pre-initialized" GNNs (Kipf & Welling, 2016). These GNNs are initialized using standard Xavier (Glorot) normal weights with a fixed seed (0). This ensures that the functional mapping of the latent confounding mechanism remains stable and reproducible across all seeds while the specific graph structures and feature values vary.

