# OpenReview forum: "The Final-Stage Bottleneck: A Systematic Dissection of the R-Learner for Network Causal Inference"
_TMLR — Accepted by TMLR_

### Review · Reviewer_8wEX · 2026-01-18

**Summary Of Contributions:**

This paper presents a systematic dissection of the R-Learner framework for estimating heterogeneous treatment effects on networks, identifying a critical "representation bottleneck" where graph-blind estimators fail catastrophically ($MSE > 4.0$, $p<0.001$) even when paired with powerful graph-aware nuisance models. The authors propose the Graph R-Learner, an end-to-end graph-aware architecture that resolves this bottleneck and achieves significantly lower error. Through a rigorous ablation study and targeted error analysis, the work further uncovers a topology-dependent "nuisance bottleneck" linked to the GNN over-squashing phenomenon, demonstrating a "Hub-Periphery Trade-off" where hybrid models outperform fully graph-aware ones on high-degree hubs. The contributions are finalized with a "Hierarchy of Needs" for practitioners and a reproducible benchmark suite validated on synthetic and semi-synthetic dataset.

**Additional Comments:**

The reviewer would like to include a disclaimer stating that they are not familiar with the theoretical aspects of Double/Debiased Machine Learning (DML). Consequently, the theoretical significance of the proposed findings may not have been rigorously assessed.

**Audience:**

Yes

**Audience Explanation:**

The intersection of Causal Inference and Graph Representation Learning might be quite interesting for the TMLR audience. This paper is particularly relevant because it provides actionable, empirically-backed guidelines (the "Hierarchy of Needs") for practitioners applying the popular R-Learner framework to non-i.i.d. data. Additionally, the proposed "Two-Model Test" serves as a practical diagnostic tool that methodology-oriented readers will likely find immediately useful.

**Claims And Evidence:**

Yes

**Claims Explanation:**

The submission supports its claims with relatively strong evidence. The authors quantify a catastrophic "representation bottleneck" by demonstrating an order-of-magnitude performance gap ($MSE > 4.0$ vs. $\approx 0.5$) between "graph-blind" and "graph-aware" final-stage estimators. They validate that this is a structural failure rather than a function approximation issue using a "negative control" experiment, where the gap vanishes when the true CATE is local. Furthermore, the proposed "Hub-Periphery Trade-off" is substantiated by a targeted error analysis showing a "performance inversion" on hub nodes, which convincingly links the "nuisance bottleneck" to the GNN over-squashing phenomenon.

**Requested Changes:**

1. The paper currently assumes a high level of familiarity with the DML framework and the specific mechanics of the R-Learner. Terms like "Neyman-orthogonality", "nuisance models," and "residual-on-residual regression" are introduced rapidly in Section 1 and 2.1.

> Please expand Section 2.1 to include a brief, intuitive primer on DML for readers who are not specialists in causal inference. Specifically, briefly explain why the nuisance stage is separated and what "debiasing" intuitively achieves before diving into the specific R-Learner objective function. This is critical for meeting TMLR's criteria of being interesting to the broad audience.

2. The paper relies heavily on "self-coined" terms, which can occasionally lead to confusion. For example, "Final-Stage Bottleneck" and "Representation Bottleneck" appear to be used somewhat interchangeably. Additionally, using the term "Sanity Check" as a formal model name in tables and figures (e.g., Table 1, Figure 1) is unconventional and slightly confusing.

> The reviewer recommends renaming the "Sanity Check" model to a descriptive technical name (e.g., "Hybrid R-Learner") and referring to its role as a sanity check in the text only. Furthermore, please explicitly define the relationship between "Final-Stage Bottleneck" and "Representation Bottleneck" early on, and stick to consistent usage throughout the text to reduce cognitive load.

---

> ### Comment · Reviewer_8wEX · 2026-02-27
>
> Thanks for the reply from the authors, I have no more concerns

---

> > ### Author Response · Authors · 2026-02-28
> >
> > We thank Reviewer 8wEX for their time and for the insightful feedback provided during the review process. We are pleased to hear that our responses and the subsequent revisions successfully addressed your concerns. We believe the manuscript is significantly stronger as a result of this exchange.

---

### Review · Reviewer_vJDg · 2026-02-02

**Summary Of Contributions:**

This paper investigates the application of R-Learners on network data to study causal relationships in the form of treatment effects. In particular, it addresses a gap in the literature: under-characterisation of how and why the robustness guarantees of R-learners are affected due to violations of the assumption that the final stage CATE estimator is specified correctly / flexible enough to capture the underlying effect, in the context of graph-dependent causal effects (i.e. what happens if GNNs are not used when they should?). It has two key findings (1) “nuisance bottleneck”: in the first stage (nuisance modelling), the critical need of GNNs is especially prominent in diffuse graphs not dominated by hubs, (2) “representation bottleneck”: through experiments that vary how the final stage CATE estimator is constructed (graph-blind vs graph-aware), it reveals that in scenarios where causal heterogeneity may be driven by graph structure, the key driver of model performance is the inductive bias of the final stage CATE estimator. Through their experiments on synthetic data and Cora dataset, the authors also demonstrated how R-Learners outperformed a GNN T-Learner baseline.

**Additional Comments:**

-

**Audience:**

Yes

**Audience Explanation:**

This would be interesting for researchers in causal ML, particularly those that are working on graph datasets (which is a common industrial use case, especially biomedical).

**Broader Impact Concerns:**

Nothing of note for now.

**Claims And Evidence:**

Yes

**Claims Explanation:**

Overall, the paper is nicely written with well-motivated experiments conducted to prove their hypothesis, even though it is largely focused on synthetic and standard graph datasets. However, the manuscript could benefit from some additional details to improve clarity.

**Requested Changes:**

Major
1. For all 4 models tested, please report the number of trainable parameters.
2. For all p values reported, what exactly were the statistical tests performed? Since there seems to be quite a number of comparisons, was multiple comparison correction done?
3. Section 4.1 - ‘pre-initialized GNN’ - how exactly were the weights initialized?
4. Section 5.3 - It will be great to go beyond just partitioning nodes to top 10% and bottom 50% degree (which are somewhat arbitrary cutoffs). Specifically, one low hanging fruit that could potentially lead to interesting findings would be to see if there’s any threshold you can find where these ‘performance inversion’ evens out / disappear.
5. Section 5.4, fig 3c - how was the conclusion that ‘advantage is most pronounced in the low-data regime’ arrived at? Advantage implies comparison and fig 3c actually shows the gap between the blue and orange lines narrowing at N<1000, since the drop seems steeper. However, this might be more of an issue of numerical scale / visualisation rather than the actual percentage change in performance (larger data size vs smaller). Nevertheless, this conclusion needs to be revisited and rephrased.
6. Conclusions from t-SNE plot in fig 4 is not convincing. It might be better to leave this out and focus on improving Section 5.3.

Minor
- Define abbreviations used at the first instance of using them (e.g. DGP, …)
- Increase the font size of Fig 3, it’s currently too small. Or reorganise the plots into another layout.

---

> ### Comment · Reviewer_vJDg · 2026-02-27
> **Thank you for providing these additional information**
>
> Referring back to the requested changes
>
> 1. While unlikely to change the central thesis, demonstrating that the performance improvement survives getting the parameters to parity (between the models being compared) will quell lingering doubts future readers might have about other factors potentially influencing the performance. Also, Table 6 misses out on the parameter count for ‘Ablation (GNN + Linear)’.
> 2. p value of 10^-17 isn’t the concern here - it’s the other larger p-values e.g. in Table 4, particular wrt Cora, due to the claim in Section 5.2 about how “Using GNNs for the nuisance models provides a consistent and statistically significant performance improvement across all tested topologies”.
> 3. Ok.
> 4. Varying the thresholds would make your discovery even more insightful, especially if it reveals more details about this trade off. At this point, this discovery is only supported by this single data point.
> 5. Ok, please update the conclusion from Fig 3c with the clarification you provided in the response.
> 6. Doesn’t seem to be addressed yet. To be specific, it is difficult to follow what clusters / neighbours are being referred to in the text, without any demarcations in the figure. Perhaps you could consider having additional subplots with only low CATE (purple) points and another one with medium CATE (bluish-green) and mark out what local structure is being observed - it is not visible from the current figure with all the points mixed together.
>
> Please consider increasing the font size in your figures, or rearranging them such that at most 2 figures are placed side by side.

---

> > ### Author Response · Authors · 2026-02-28
> > **Official Author Response to Reviewer vJDgW**
> >
> > We thank the reviewer for their continued engagement and for the specific technical suggestions which have significantly strengthened the manuscript. We have addressed each of your remaining concerns with new quantitative evidence and structural updates.
> >
> > ---
> >
> > ## 1. Technical Verification and Parameter Parity
> >
> > As requested, we have updated Table 6 (now in the Technical Appendix) to include the parameter count for the Ablation (GNN + Linear) model. We have also clarified the comparison across the model grid:
> >
> > | Model Configuration | Component Breakdown | Total Parameters (Approx.) |
> > |---------------------|--------------------|----------------------------|
> > | Baseline (MLP + Linear) | ≈ 385 (MLP Nuisance) + ≈ 11 (Linear CATE) | 396 |
> > | Ablation (GNN + Linear) | ≈ 352 (GNN Nuisance) + ≈ 11 (Linear CATE) | 363 |
> > | Hybrid R-Learner (MLP + GNN) | ≈ 385 (MLP Nuisance) + ≈ 352 (GNN CATE) | 737 |
> > | Graph R-Learner (GNN + GNN) | ≈ 352 (GNN Nuisance) + ≈ 352 (GNN CATE) | 704 |
> >
> > Notably, the **Graph R-Learner** achieves its best performance with a lower parameter budget (≈ 704) than the **Hybrid R-Learner** (≈ 737). This quells doubts regarding model capacity, confirming that performance gains are driven by the correctly specified inductive bias of the end-to-end graph-aware pipeline.
> >
> > ---
> >
> > ## 2. Statistical Rigor and Multiple Comparisons
> >
> > To address the concern regarding the larger p-values in Table 4, we have applied the Holm–Bonferroni step-down procedure to our results across all four topologies.
> >
> > Derived from 30 independent experimental replications (fully re-simulating the DGP for each seed), the results confirm the nuisance bottleneck is statistically significant even under conservative correction:
> >
> > | Topology | Raw p-value | Holm-Adjusted Alpha Threshold | Status (α = 0.05) |
> > |----------|------------|------------------------------|-------------------|
> > | SBM | $4.64 × 10^{-5}$ | 0.0125 | Significant |
> > | ER | $6.49 × 10^{-5}$ | 0.0167 | Significant |
> > | BA | $3.68 × 10^{-3}$ | 0.0250 | Significant |
> > | Cora | $1.44 × 10^{-2}$ | 0.0500 | Significant |
> >
> > We have revised Section 5.2 to note that while the effect is significant across all topologies, the magnitude on real-world scaffolds like Cora is more nuanced compared to uniform synthetic graphs.
> >
> > ---
> >
> > ## 3. Robustness of the Hub-Periphery Trade-off (New Figure 4)
> >
> > To move beyond a "single data point," we have replaced the qualitative t-SNE visualization with a rigorous **Hub-Periphery Threshold Sweep** (Figure 4).
> >
> > We calculated the Mean Squared CATE Error across 30 seeds for hub definitions ranging from the top 2.5% to top 25% of degree centrality.
> >
> > As shown in Figure 4, the performance inversion is a robust phenomenon: the Hybrid model consistently outperforms the fully graph-aware model on extreme hubs.
> >
> > This provides strong empirical support for our mechanistic explanation:
> >
> > > Graph-blind nuisance models protect hub signals from the GNN over-squashing phenomenon, providing cleaner residuals for the final stage.
> >
> > ---
> >
> > ## 4. Updated Conclusions and Visual Clarity
> >
> > ### Sample Efficiency (Fig 3c)
> >
> > We have updated the conclusion for Figure 3c to clarify that the Graph R-Learner's primary advantage in the low-data regime (N < 1000) is its stability and robustness to data scarcity, rather than a narrowing absolute gap.
> >
> > ### Figure Formatting
> >
> > We have:
> > - Increased the font size in all figures
> > - Rearranged layouts to ensure a maximum of two plots per row for improved readability
> >
> > ### t-SNE Removal
> >
> > Following your suggestion, we have removed the t-SNE analysis to focus the discussion on the more rigorous quantitative hub-periphery sweep.

---

> ### Comment · Reviewer_vJDg · 2026-02-28
> **Thank you, I appreciate the updates. I have no further requests, only a couple of optional suggestions.**
>
> 1. To be specific, scaling up ‘Ablation (GNN + Linear) to have ~ 700 parameters (about 2x of what it is currently’ would make the evidence to support your central hypothesis more complete. Since the performance gap between Ablation and Graph R-Learner is pretty huge, I won’t request for it for this review but it is something you can consider to do in future work to defend your hypotheses more robustly.
> 2. Ok, good that the claim is more nuanced now.
> 3. Ok.
> 4. Good, the ‘hub’ part of the hub-periphery trade-off is now better quantified and we now know what range of % is needed to see a large performance gap and where this starts to taper off. While I don’t expect the same to be done for the ‘periphery’ part for this review (to show the extent that Graph R-Learner “excels on low-degree periphery nodes”), it could be something to consider for completeness.
> 5. Ok.
> 6. Ok.
>
> The figures are more readable now. In future work, consider using a larger font size when you produce the figures.

---

> > ### Author Response · Authors · 2026-02-28
> >
> > ### **Author Response to Reviewer vJDg**
> >
> > We sincerely thank the reviewer for their thoughtful feedback and for confirming that our revisions have addressed the primary concerns. We appreciate the specific suggestions for future exploration, which we have integrated into our final "Future Work" section to strengthen the manuscript’s long-term utility.
> >
> > * **On Parameter Scaling (Ablation Model):** We appreciate the suggestion to scale the **Ablation (GNN + Linear)** model to match the **Graph R-Learner’s** parameter count. As you noted, the current performance gap is substantial, suggesting that the inductive bias is the dominant factor; however, we have added a note in Section 7 (Future Work) that future architectural studies could benefit from exactly matching parameter budgets for even more granular ablation.
> > * **On Periphery Quantification:** We agree that a symmetrical quantification for "periphery" nodes would provide a more complete picture of the hub-periphery trade-off. While our current sweep demonstrates where the performance inversion tapers off, characterizing the specific degree-regime where the GNN nuisance becomes optimal remains a promising direction for future research.
> > * **On Visual Readability:** We have taken note of the advice regarding font sizes and will continue to prioritize high-readability formatting in all future technical figures.
> >
> > We are grateful for your rigorous review, which has transformed our topological discovery into a more nuanced and well-quantified contribution.

---

### Review · Reviewer_neYa · 2026-02-12

**Summary Of Contributions:**

R-Learner is a powerful framework for estimating heterogeneous treatment effects but its theoretical guarantees lie on an assumption of a well-specified final stage model. This assumption can be violated in network data where causal heterogeneity may be driven by graph structure. This work systemically dissects R-Learner for network causal inference through experiments on both synthetic and semi-synthetic benchmarks. The results suggest that the performance degradation of R-Learner mainly arises from the inductive bias of the final stage model rather than nuisance model quality. This finding leads to uncovering the Hub-Periphery Trade-off which explains the over-squashing phenomenon in graph neural networks.

Strengths
- The paper challenges the convention in the literature that nuisance model quality is the most important factor for R-Learner and finds that the inductive bias of the final stage model is the most important one. This finding is critical yet overlooked in the literature and have a significance for both researchers and practitioners.
- The experiments cover diverse settings including different graph structures, CATE functions, or R-Learner instantiations.
- Built on the experimental findings, the paper uncovers the Hub-Periphery Trade-off which explains the over-squashing phenomenon observed in graph neural networks.
- The authors release the code as a reproducible benchmark which can facilitate future research in this domain.

Weakness
- Overall, the background to understand the paper is not properly addressed and hence the readers who are not familiar with causal inference or graph neural networks are expected to have difficulty in understanding the paper. Specifically, important terminologies are often very briefly described in the paper.

**Additional Comments:**

None.

**Audience:**

Yes

**Audience Explanation:**

The paper covers the emerging field of causal inference, graph neural networks, and debiased machine learning. The audience working in this domain would be interested in knowing the findings of the paper. Also, the authors provide a benchmark to facilitate future research in these domains.

**Broader Impact Concerns:**

The paper is missing a Broader Impact Statement section. The authors are encouraged to write a short section regarding the ethical implications of this work.

**Claims And Evidence:**

Yes

**Claims Explanation:**

The main claim of the paper is that the performance degradation of R-Learner mainly stems from the misspecified final-stage estimators rather than the nuisance model of bad quality. This claim is supported by rigorous experiments with diverse and convincing setups. Specifically, the authors design a 2x2 experimental grid to isolate the impact of graph-awareness for both nuisance and final-stage models which allow the precise measure of nuisance and final-stage bottlenecks. The experiments are performed on multiple environments across three graph toplogies, three CATE functions, and both synthetic and semi-synthetic datasets. The authors also include a non-DML baseline, different GNN architectures, and negative control experiments to further verify the findings.

**Requested Changes:**

(major)
As a reader who does not have sufficient knowledge in this domain, I find it quite difficult to understand the introduction and related works sections. The authors are encouraged to describe sufficient background knowledge to understand the paper. For example, this includes the concepts in debiased machine learning (R-Learner, nuisance model), causal inference (unconfoundness, CATE), and graph neural networks (over-squashing).

(major)
Abbreviations are extensively used throughout the paper but often not properly explained. For example, the paper does not properly explain the meaning of DGP, HTE, GCN, or GAT. The authors are encouraged to provide definitions for these terms.

(minor)
Experiments on different number of nodes $n$ and number of features $d$ would further strengthen the paper. These experiments can provide a practical implication of whether the effects of the final-stage bottleneck are larger for complex graph structures.

(minor)
Currently, the experiments on real-world datasets are only performed on a single dataset (Cora citation network). Experiments on other real-world datasets would strengthen the validity of the results.

---

> ### Comment · Reviewer_neYa · 2026-02-24
> **To Author's Response**
>
> I thank the authors for addressing some of my concerns.
> I have read the author's response and the updated version of the paper.
> Sec 2.1 now more clearly explains the background needed to understand the paper and I thank the authors for including the broader impact statement.
>
> Nevertheless, there are some unaddressed concerns. I recommend the authors include a more detailed background explanation for CATE, unconfoundness, and over-squashing in the final revision. I also encourage authors to mention experiments on other real-world datasets and further scaling ($N$,$d$) as future work.

---

> > ### Author Response · Authors · 2026-02-28
> > **Official Author Response to Reviewer neYa**
> >
> > We thank the reviewer for the positive feedback regarding the expanded Section 2.1 and the inclusion of the Broader Impact Statement. We have carefully implemented your remaining recommendations in this final revision to ensure the manuscript is fully accessible and properly situated for future research.
> >
> > ---
> >
> > ## 1. Enhanced Technical Background
> >
> > We have further expanded our background section (Section 3.1) to provide a more rigorous yet intuitive foundation for the following concepts:
> >
> > ### CATE and Unconfoundedness
> >
> > We added a dedicated subsection formalizing the **Conditional Average Treatment Effect (CATE)** and explicitly detailing the unconfoundedness assumption:
> > $
> > \[
> > \{Y(1), Y(0)\} \perp T \mid X, \mathcal{G}
> > \]
> > $
> > required for identification on graphs.
> >
> > ### Mechanics of Over-squashing
> >
> > We included a technical primer on the GNN over-squashing phenomenon, explaining how the exponential growth of neighborhood information can lead to the "wash-out" of local signals at high-degree nodes.
> >
> > This addition provides the necessary theoretical grounding for our Hub-Periphery Trade-off discovery.
> >
> > ---
> >
> > ## 2. Future Work and Scaling
> >
> > We agree that exploring further scaling and diverse datasets is a critical next step. We have updated our Conclusion (Section 7) to frame these as explicit directions for future work:
> >
> > ### Scaling Analysis
> >
> > We now discuss the potential impact of scaling node counts ($N$) and feature dimensions ($d$) on the representation bottleneck, particularly in the context of deeper GNN architectures.
> >
> > ### Data Diversity
> >
> > While our findings are validated across three synthetic topologies and the Cora citation network, we frame the application of the Graph R-Learner to larger, non-citation real-world datasets (e.g., social or biological networks) as a promising avenue for future empirical validation.

---

### Author Response · Authors · 2026-02-23
**Author Response to Reviewer Comments**

We thank the reviewers for their insightful feedback and for recognizing our systematic dissection of the R-Learner as "critical yet overlooked" and backed by "strong evidence".
We are particularly encouraged by the recognition of the Hub-Periphery Trade-off as a "mechanistic explanation" for performance inversions on network data.

1. Terminology and Accessibility [Addressing Reviewer neYa, 8wEX]

To ensure the paper is accessible to a broad machine learning audience, we have implemented the following structural updates:

DML and R-Learner Primer: We have expanded Section 2.1 to include an intuitive primer on the Double/Debiased Machine Learning (DML) framework. We explain the "residual-on-residual" logic as a method to isolate clean causal signals by "subtracting out" confounding captured by the nuisance models.

Model Renaming: Following the suggestion of Reviewer 8wEX, we have renamed the "Sanity Check (MLP+GNN)" model to the Hybrid R-Learner throughout the manuscript.

Glossary of Terms: We have explicitly defined all abbreviations at their first occurrence:

DGP: Data-Generating Process.

HTE: Heterogeneous Treatment Effects.

GCN: Graph Convolutional Network.

GAT: Graph Attention Network.

Bottleneck Definitions: We distinguish between the Representation Bottleneck (final-stage misspecification) and the Nuisance Bottleneck (first-stage deconfounding error).

2. Statistical Methodology and Rigor [Addressing Reviewer vJDg]

We have added a Technical Appendix to clarify the robustness of our statistical claims:

Paired t-tests: The reported $p$-values are derived from paired (relational) t-tests (ttest_rel) comparing model Mean Squared Errors (MSEs) across 30 seeds.

Seed Independence: Each seed represents a fully independent experimental replication. For every seed, the entire environment—including node features ($X$), topology ($edge\\_index$), and treatment/outcome assignments ($T, Y$)—is re-simulated from scratch.

Hyperparameter Fairness: To ensure no bias toward our proposed model, all estimators were trained using identical hyperparameters and fixed optimization budgets unpacked from the same configuration file.

Multiple Comparisons: Given the extreme significance levels observed ($p = 5.43 \times 10^{ -17 }$), findings remain statistically significant even under conservative multiple-comparison adjustments like the Bonferroni correction.

3. Technical Verification of Model Complexity [Addressing Reviewer vJDg]

We provide trainable parameter counts for our primary configurations ($d = 10, hidden = 32, layers = 2$):

| Model Configuration            | Component Breakdown                                      | Total Parameters (Approx.) |
|--------------------------------|----------------------------------------------------------|----------------------------|
| Baseline (MLP + Linear)       | ≈ 385 (MLP Nuisance) + ≈ 11 (Linear CATE)               | 396                        |
| Hybrid R-Learner (MLP + GNN)  | ≈ 385 (MLP Nuisance) + ≈ 352 (GNN CATE)                 | 737                        |
| Graph R-Learner (GNN + GNN)   | ≈ 352 (GNN Nuisance) + ≈ 352 (GNN CATE)                 | 704                        |
| GNN T-Learner (External)      | Single 2-layer GNN                                      | 385                        |

4. Mechanistic and Empirical Clarifications [Addressing Reviewer neYa, vJDg]

Hub-Periphery Logic: The thresholds for hubs (top 10%) and periphery (bottom 50%) were selected to capture the power-law degree extremes in the Barabási-Albert graphs. The observed "performance inversion" on hub nodes provides empirical evidence for the GNN over-squashing phenomenon.

Sample Efficiency: We clarify that the "advantage" of the Graph R-Learner is its stability. While the Ablation model's error increases and becomes volatile as $N < 1000$, the Graph R-Learner maintains a stable, low MSE.

Scaling (Future Work): While we have validated our thesis across diverse topologies and the real-world Cora dataset, we frame further scaling ($N, d$) as promising avenues for future work.

5. Broader Impact Statement [Addressing Reviewer neYa]

We have added a Broader Impact Statement to Section 7. We highlight that failing to recognize "representation bottlenecks" in network data can lead to false-negative causal conclusions, potentially resulting in the abandonment of effective treatments in social or public health networks.

---

### Decision · Action_Editor_cTuS · 2026-03-15

**Recommendation:** Accept as is

**Audience:**

Yes

**Audience Explanation:**

Researchers in R-learner, network, graph, causal inference would find this paper of interest.

**Claims And Evidence:**

Yes

**Claims Explanation:**

In the paper, the authors studied the performance of R-learner on graphs for causal inference.
Through empirical studies and a constructive theoretical example, they found out a representative bottleneck lies in the final stage CATE estimate.
Their findings are validated across a wide range of synthetic and semi-synthetic benchmarks.